# Chronic Diseases and Associated Factors among Older Adults in Loja, Ecuador

**DOI:** 10.3390/ijerph17114009

**Published:** 2020-06-04

**Authors:** Patricia Bonilla-Sierra, Ana-Magdalena Vargas-Martínez, Viviana Davalos-Batallas, Fatima Leon-Larios, Maria-de-las-Mercedes Lomas-Campos

**Affiliations:** 1Departamento de Ciencias de la Salud, Universidad Técnica Particular de Loja, Loja 110107, Ecuador; pbonilla65@utpl.edu.ec (P.B.-S.); vddavalos@utpl.edu.ec (V.D.-B.); 2Nursing Department, Faculty of Nursing, Physiotherapy and Podiatry, University of Seville, 41009 Seville, Spain; mlomas@us.es

**Keywords:** chronic disease, symptoms, dependence, performance, depression, Ecuador

## Abstract

(1) Background: This study aimed to explore the symptoms, functional status, and depression in patients with chronic diseases in Loja, Ecuador. (2) Methods: A cross-sectional study was carried out with patients over 60 years old having at least one chronic disease and cared for in healthcare centers of the Health Ministry of Ecuador or living in associated geriatric centers. (3) Results: The sample comprised 283 patients with a mean age of 76.56 (SD 7.76) years. The most prevalent chronic diseases were chronic obstructive pulmonary disease, followed by arterial hypertension and diabetes. Patients with a joint disease had the worst scores for the majority of the symptoms assessed with the Edmonton Scale. Cancer, dementia, and arterial hypertension contributed the most to the dependence levels assessed with the Barthel Index. Dementia contributed the most to the poor performance status evaluated with the Karnofsky Performance Status. Cancer and diabetes contributed the most to depression. Patients with a higher number of chronic diseases reported worse functional status. (4) Conclusions: Targeted interventions to address symptoms, functional status, and depression in patients with chronic diseases are needed.

## 1. Introduction

Non-Communicable Diseases (NCDs), also known as chronic diseases, are the most important cause of mortality worldwide and contribute 71% of all deaths globally. Cardiovascular diseases, cancers, chronic respiratory diseases, and diabetes are the four main types, accounting for over 80% of all the premature deaths [1]. Notably, 78% of the global deaths due to NCDs occur in low- and middle-income countries [2].

In Ecuador, the main source of data on chronic diseases is provided by the National Survey of Health and Nutrition (*Encuesta Nacional de Salud y Nutrición*, ENSANUT). Regarding mortality, ENSANUT data show that ischemic heart disease, diabetes mellitus, hypertensive disease, cerebrovascular disease, and chronic diseases of the lower respiratory tract are among the top ten causes of death representing 31.2% of all deaths [3,4,5]. All types of cancers have been the second leading cause of death in Ecuador in recent years [6]. However, these data may be underreported. In 2011, the Pan American Health Organization (PAHO) estimated an underreporting of 16.7% [5] for the country, and this could also be reflected in the proportion of deaths due to undefined causes, which was 6.5% in 2018 [7].

In the annual epidemiological surveillance, a national rate of NCDs in Ecuador in 2017 was 2740.93 NCD cases per 100,000 inhabitants, with a total of 485,845 cases and significant differences among the regions [5,8]. Furthermore, the prevalence rates of the main chronic NCDs have been increasing over the years in both genders and are also higher as age increases [9].

NCDs cause functional problems and present personal, economic, and social burdens. Moreover, symptoms, such as pain, may affect the psychological condition of the patients, and depression is one of the most frequent emotional reactions [10,11,12]. For all these reasons, knowledge about the impact of NCDs on symptoms, functional status, and depression is needed to allow health professionals to improve care and help both patients and family caregivers.

We have carried out this study in the absence of published similar papers in Ecuador. The main objective was to describe symptoms, functional status, and depression in patients with NCDs. The specific objectives were as follows:(a)To measure symptoms related to different NCDs;(b)To study the existing associations between the functional status (i.e., dependence and performance levels) and the different NCDs, as well as the number of NCDs;(c)To analyze the existing associations between depression and the different NCDs, as well as the number of NCDs.

## 2. Materials and Methods

### 2.1. Study Design and Settings

This was a cross-sectional epidemiological study that was conducted in Loja, Ecuador, from January to December 2019. The study adhered to the STROBE (STrengthening the Reporting of OBservational studies in Epidemiology) statement for observational studies [13]. Loja is an area of Ecuador with a total approximate population of 500,000 and is located in the South of the country, near Peru. Loja is one of the 24 provinces of Ecuador, and it ranks as the 10th most populous area in the country. It is composed of 16 “cantones” (villages).

### 2.2. Sampling and Inclusion

Patients who were cared for in healthcare centers of the Health Ministry of Ecuador or were living in a geriatric center associated with these healthcare centers were contacted. A convenience sampling was used; we contacted patients who (1) attended an outpatient appointment in the clinic to follow up their chronic problem (2) during home visits and (3) during geriatric institution visits. Patients were sampled from January to December 2019.

The study inclusion criteria were as follows: (1) individuals who had one or more self-reported chronic diseases or a risk factor to develop another chronic condition, such as obesity; (2) over 60 years of age; and (3) willing to cooperate during the interview. In the case of those patients who, due to their health conditions, were unable to communicate, the main caregiver was asked to answer the questions. Patients aged 60 years old and over were included because the World Health Organisation (WHO) determines older population at this age. At present, there is no United Nations (UN) standard numerical criterion, but the UN agreed cutoff point is 60 years old to refer to the older population.

### 2.3. Measures

The participants were asked to complete face-to-face questionnaires, including sociodemographic information and validated tools to measure different health conditions. The measurements included Body Mass Index (BMI), which was codified in following categories: underweight; normal weight; overweight; and obesity classes I, II, and III. Several symptoms were measured by means of the Edmonton Symptom Assessment Scale (ESAS), which uses a Likert-type scale (0–10) to score each symptom, where “0” is absence of symptoms, and “10” is the worst situation. The functional characteristics of the subjects were analyzed through different scales or indexes according to the Clinical Guidelines for Geronto-Geriatric Primary Health Care for Older Adults and to the Guidelines for Palliative Primary Health Care published by the Public Health Ministry in Ecuador: Barthel Index, Detection of Emotional Distress (DED) scale, Katz Index, Karnofsky Performance Status (KPS), Lawton and Brody Instrumental Activities of Daily Living (IADL) Scale, The Modified Mini-Mental State Test, Portable Functional Assessment Questionnaire (Pfeiffer), and The Modified Geriatric Depression Scale of Yesavage [14]. The KPS score was recoded as “1” for those who scored between 85 and 100, and as “0” for those who scored less than 85. In accordance with the Clinical Guidelines for Geronto-Geriatric Primary Health Care for Older Adults published by Ecuador’s Public Health Ministry [14], the modified Katz Index consists of 8 items. Each item is scored on three levels: 0 (dependent), 1 (moderate dependence), and 2 (independent). Then, the total score can range from 0 to 16. Several ways of the total score interpretation have been proposed [15,16,17,18,19,20,21,22,23,24]. Given the agreement gap regarding the classification of the modified index used in Ecuador and since the guidelines used do not specify any global classification of this scale, in this study, the total score of the Katz Index was recoded into three categories: severe dependence (0–7 points), moderate dependence (8–15 points), and independence (16 points).

### 2.4. Statistical Analysis

The demographic characteristics of the patients were described as means ± standard deviations or as n (%) of the total. In addition, *t*-tests, Pearson’s *χ*^2^, and Fisher tests, depending on the type of variable and its answer’s frequency, were used to compare characteristics by gender. An analysis of Visual Analogue Scale (VAS) means for symptoms related to different chronic diseases, as well as a comparative analysis of functional characteristics in patients with different numbers of chronic diseases were carried out. For this, Pearson’s *χ*^2^ and Kruskal–Wallis *χ*^2^ were used depending on the type of variable and its normality. The associations between dependence levels measured with the Barthel Index and different chronic diseases controlled by different covariates (age, gender, Body Mass Index (BMI)) were studied using multinomial logistic regression analysis. Similarly, the associations between depression (measured through The Modified Geriatric Depression Scale of Yesavage) and chronic diseases controlled by different covariates (age, gender, marital status, social risk situation, dependence level, disability level, abilities, and needs) were also studied using multinomial logistic regression analysis. The associations between functional status measured with the KPS Index and different chronic diseases were studied using bivariate logistic regression analysis, after re-coding the KPS Index as a dichotomous variable. Some variables, such as age, were centered using medians, because a value of 0 was not included in these variables in the sample studied. Data was entered into R software (R-3.6.3 version) for statistical analysis. A *p* value < 0.05 was accepted as statistically significant.

### 2.5. Ethical Considerations

The institutional Review Board of the Economic and Social Inclusion Ministry of Ecuador approved this research (Code 2019_2587). Written consent was required for all the participants; those that were illiterate were asked to give their verbal consent. Confidentiality and anonymity were guaranteed.

## 3. Results

### Demographic Characteristics

A total of 283 patients comprised the sample; the mean age was 76.56 (SD 7.76) years old. Over half (54.06%) were men, the majority was Catholic (97.88%), 45.94% were married, and 29.33% were widows/widowers. Nearly half of the patients had normal weight (45.76%), followed by those who were overweight (31.37%) and by those who had class I obesity (14.76%). A large proportion of individuals had visual disorders (36.75%). Regarding chronic diseases, the most prevalent one was chronic obstructive pulmonary disease (COPD) (58.66%), followed by arterial hypertension (53.71%) and diabetes (28.27%) (see Table 1).

Concerning the symptoms, the statistically significant difference reporting pain was marked between those who had arterial hypertension, COPD, chronic liver insufficiency, diabetes, or joint disease and those who did not have these conditions. Statistically significant difference reporting the majority of other symptoms was marked between those who had a joint disease compared to those who did not have it. In addition, those with a joint disease registered the highest scores for constipation, difficulty sleeping, dyspnea, hyporexia, nausea, and sleepiness (see Table 2).

In the comparative analysis of the functional characteristics of patients with different numbers of chronic diseases, statistically significant differences were found in relation to the dependence level measured with the Barthel Index, the disability levels measured with the Katz Index, the performance levels measured with the KPS and the cognitive impairment levels measured with the Portable Functional Assessment Questionnaire. All these measures indicated worse functional status in patients with higher number of chronic diseases. Another significant finding was that a higher percentage of depression was found among those who reported having two chronic diseases (see Table 3).

Then, analysis of the association between dependence level and different chronic diseases through a univariate logistic regression model showed those who reported having visual disorders, cancer, and dementia presented a higher risk of reaching severe or moderate dependence than those who did not have those conditions. Moreover, those who were obese had a lower risk of developing severe or moderate dependence than those who were classified as normal weight. In addition, an increase in the number of chronic diseases resulted in an increased risk of developing severe or moderate dependence. Those with three or more chronic diseases had a 4.98 times higher risk of developing severe or moderate total dependence than those who reported none or one chronic disease (see Table 4, crude ORs). In the multinomial logistic regression model, cancer, dementia, and arterial hypertension were significant risk factors for developing severe or moderate dependence. In addition, obesity turned out to be a protective factor to develop severe or moderate dependence compared to normal weight (see Table 4, adjusted ORs).

When analyzing the association between performance measures with the KPS Index and different chronic diseases through a univariate logistic regression model, those with visual disorders, cancer, dementia, and arterial hypertension had more risk to score less than 85 points in the KPS Index (i.e., to have worse performance) than those who did not have those diseases. Additionally, a one-year increase in age increased by 1.07 times the risk to score less than 85 points in the KPS Index. In addition, being a man turned out to be a protective factor to score less than 85 points in the KPS Index, and a higher number of chronic diseases turned out to be a risk factor to score less than 85 points in the KPS Index (See Table 5, Crude ORs). In the adjusted model, the associations with dementia, age, and gender remained significant (see Table 5, adjusted ORs).

In relation to the associations between depression and different chronic diseases, a univariate regression model reported that those with visual disorders, cancer, diabetes, a social risk situation (Gijón Scale), dependence (Barthel Index), <85 points in the KPS Index, and semi-dependence or dependence (Lawton and Brody IADL scale) had a higher risk of developing depression or probable depression than those who did not have these diseases and conditions (see Table 6, Crude ORs). In the multivariate regression model, those who had cancer, diabetes, and those who reported a social risk situation (Gijón Scale) had a higher risk of developing depression or probable depression than those who did not have these diseases and conditions. In addition, those who scored less than 85 points in the KPS Index presented a 2.84 times higher risk than those who scored 85 or more points. (see Table 6, Adjusted ORs).

## 4. Discussion

This is one of the few studies published in Ecuador that provide information about patients who suffer from one or more chronic diseases and their symptoms, functional situation, and depression level. The main results of this study provide a picture of the situation of chronicity in an area of Loja, Ecuador. Thus, the most prevalent chronic disease was COPD, followed by arterial hypertension and diabetes. The most common symptom was pain. Patients who suffered from a joint disease had more symptoms in general than those who did not have this disease. Cancer, dementia, and arterial hypertension contributed the most to the dependence levels; dementia contributed the most to the poor performance status; and cancer and diabetes contributed the most to depression. Patients with a higher number of chronic diseases reported worse functional status.

The prevalence of non-communicable diseases has doubled in low- and middle- income countries in recent decades [25]; coping with this is challenging particularly in Latin American and Caribbean nations [26]. Morbidity associated with non-communicable diseases mainly affects older people, aged 60 years old and more [27]. This burden is higher in low- and middle- income regions such as Ecuador where the most predominant diseases include chronic respiratory diseases, neoplasia, cardiovascular diseases, musculoskeletal diseases, and neurological and mental disorders [28,29]. Chronic disease in older people is a complex issue, as Maresova et al., (2019) suggested, and early recognition of disability helps program interventions to maintain activities to promote daily living independence [30].

COPD is a major cause of chronic morbidity throughout the world, even in Ecuador. This pathology was shown to be more frequent in females [31], also in studies carried out in low-income countries [32,33]. Males on the other hand experience higher rates of neoplasms and cardiovascular diseases than women [34].

Chronic pain is the most common symptom in older people with chronic diseases, and this is a major healthcare problem worldwide that needs to be taken into account [35]. Results reported by other studies suggest that people who suffer from chronic pain show less ability to perform daily life activities, and this increases significantly with age [11,12,31]. This finding is consistent with our study, where we observed a relation between the presence of chronic pain and dependence.

In this study, patients with cancer had a worse functional situation than those who did not have cancer. Similarly, patients with dementia and arterial hypertension were found to report worse functionality than patients with other different chronic diseases. This may contribute to decreasing the health-related quality of life of older adults [36], so it is important to promote interventions to provide support for people with cancer to participate in everyday activities [37]. Previous studies have shown that these activities contribute to better patient outcomes [38]. With regard to dementia, it is expected that, when dementia is diagnosed, the level of dependence is higher [39]. Additionally, our data align with the works by Costa Filho (2018) and Li (2019), showing that being 75 years old or older increased the risk of significantly moderate and severe dependence [40,41].

The participants in our study who were over 75 years old and had two or more chronic diseases could be considered fragile patients. There is a reciprocal interaction between depression and frailty in older adults [42]. Geriatric depression is a major public health problem and has an important effect on the health of patients with a medical problem. We observed that participants with more than one chronic disease showed higher scores of depression measured with the Yesavage scale. This finding is in line with the systematic review published by Read et al., (2017), who concluded that the presence of two or more chronic conditions increased the likelihood of depression [43]. Our study revealed no difference by gender in terms of geriatric depression; however, other studies demonstrated that a slight difference does exist [43,44]. If we take into account the social risk of the patients, we can assert that those with a higher social risk measured with the Gijón Scale can easily develop geriatric depression and a higher risk of premature mortality, as some authors concluded [45,46,47].

Patients with numerous comorbidities require management to improve their situation and avoid worsening of the disease. Actually, an approach where decisions are made in collaboration with the patients would be the best choice, as this is in favor of embracing autonomy. It is necessary to understand how to encourage patients and healthcare providers to engage in effective interactions in chronic care management in order to get better outcomes [48].

As a limitation, we must state that these data are from only one region of Ecuador and that they may not be representative of the whole country. Besides, the participants were recruited using a convenience sampling approach, and those who are not used to following up their chronic condition are missed in our sample. On the other hand, the sample was large, which may outweigh some shortcomings of nonprobability sampling [49]. Furthermore, our data align with the national statistics, except that more men than women were recruited in our study, while women are more numerous than men in Ecuador. Another limitation is that we are not able to attribute the causality between chronic diseases and symptom, functionality, and depression since this is a cross-sectional study. Nevertheless, since there is not much research in this field in the country, this study sheds light on the situation of chronicity and morbidity in Ecuador, particularly in the Loja area.

## 5. Conclusions

It is necessary to develop strategies to address symptoms, functionality, and depression in patients with chronic diseases in Ecuador. To do that, targeting of specific disease interventions is needed.

## Figures and Tables

**Table 1 ijerph-17-04009-t001:** Characteristics of Subjects (n = 283).

Variables	Total Sample n = 283	Women n = 130	Men n = 153
***Sociodemographic***			
**Age**	76.56 (7.76)	75.38 (7.87)	77.56 (7.55) **
**Marital status**			
Married	130 (45.94)	58 (44.6)	72 (47.1)
Divorced	8 (2.83)	4 (3.1)	4 (2.6)
Unmarried/Single	62 (21.91)	18 (13.8)	44 (28.8)
Widowed	83 (29.33)	50 (38.5)	33 (21.6)
**Gijon’s social-familial evaluation scale**			
Social problem (>10)	68 (24.46)	23 (17.8)	45 (30.2)
Social risk situation (8–9)	81 (29.14)	46 (35.7)	35 (23.5)
Normal social situation (<8)	129 (46.40)	60 (46.5)	69 (46.3)
***Health disorders***			
**BMI (Body Mass Index)**			
Underweight (<18.5 kg/m^2^)	17 (6.27)	6 (4.7)	11 (7.6)
Normal weight (18.5–24.9 kg/m^2^)	124 (45.76)	60 (47.2)	64 (44.4)
Overweight (25–29.9 kg/m^2^)	85 (31.37)	34 (26.8)	51 (35.4)
Class I obesity (30–34.9 kg/m^2^)	40 (14.76)	25 (19.7)	15 (10.4)
Class II obesity (35–39.9 kg/m^2^)	1 (0.37)	1 (0.8)	0 (0)
Class III obesity (≥40 kg/m^2^)	4 (1.48)	1 (0.8)	3 (2.1)
**Hearing disorders** (yes)	9 (3.18)	4 (3.1)	5 (3.3)
**Visual disorders** (yes)	104 (36.75)	44 (33.8)	60 (39.2)
**Pressure Ulcer** (yes)	17 (6.01)	8 (6.2)	9 (5.9)
***Chronic diseases*** (yes)			
**Arterial Hypertension**	152 (53.71)	76 (58.5)	76 (49.7)
**Cancer**	11 (3.89)	5 (3.8)	6 (3.9)
**Cardiovascular Disease**	15 (5.3)	9 (6.9)	6 (3.9)
**COPD**	166 (58.66)	80 (61.5)	86 (56.2)
**Chronic Heart Disease**	5 (1.77)	3 (2.3)	2 (1.3)
**Chronic Liver Insufficiency**	17 (6.01)	6 (4.6)	11 (7.2)
**Chronic Renal Insufficiency**	22 (7.77)	11 (8.5)	11 (8.5)
**Dementia**	15 (5.3)	9 (6.9)	6 (3.9)
**Diabetes**	80 (28.27)	36 (27.7)	44 (28.8)
**Joint disease**	18 (6.36)	9 (6.9)	9 (5.9)
***Edmonton Symptom Assessment Scale (ESAS)***			
**Pain** (yes)	142 (50.18)	59 (45.4)	83 (54.2)
**Anxiety**	2.25 (2.27)	2.49 (2.35)	2.04 (2.18) *
**Constipation**	1.43 (2.15)	1.68 (2.41)	1.22 (1.88) *
**Difficulty sleeping**	2.57 (2.59)	2.87 (2.68)	2.31 (2.49) *
**Dyspnoea**	0.84 (1.41)	0.91 (1.37)	0.78 (1.43)
**Fatigue**	2.52 (2.26)	2.80 (2.28)	2.29 (2.22) *
**Hyporexia**	0.92 (1.56)	1.05 (1.60)	0.81 (1.52)
**Nausea**	0.66 (1.28)	0.83 (1.41)	0.52 (1.15) **
**Sleepiness**	2.46 (2.35)	2.44 (2.41)	2.47 (2.30)
***Functional characteristics***			
**Barthel index**			
Severe dependence (20–35)	31 (10.95)	16 (12.3)	15 (9.8)
Moderate dependence (40–55)	31 (10.95)	15 (11.5)	16 (10.5)
Slight dependence (>60)	128(45.23)	66 (50.8)	62 (40.5)
Independence (100)	93 (32.86)	33 (25.4)	60 (39.2)
**Detection of Emotional Distress (DED) scale**			
With emotional distress (≥9)	123 (45.9)	57 (46.3)	66 (45.5)
Without emotional distress (<9)	145 (54.1)	66 (53.7)	79 (54.5)
**Katz Index**			
Severe Dependence (0–7)	63 (22.26)	32 (24.6)	31 (20.3)
Moderate dependence (8–15)	107 (37.81)	46 (35.4)	61 (39.9)
Independence (16)	113 (39.93)	52 (40.0)	61 (39.9)
**Karnofsky Performance Status (KPS)** (100 = normal to 0 = dead)	78.06 (20.44)	75.62 (20.76)	80.13 (20.00) *
**Lawton and Brody Instrumental Activities of Daily Living (IADL) Scale**			
Dependence (>20)	44 (15.60)	21 (16.2)	23 (15.1)
Semi-dependence (8–20)	159 (56.38)	72 (55.4)	87 (57.2)
Independence (<8)	79 (28.01)	37 (28.5)	42 (27.6)
**The Modified Mini-Mental State Test**			
Cognitive deficit (<13)	67 (23.67)	32 (24.6)	35 (22.9)
Without cognitive deficit (14–19)	216 (76.33)	98 (75.4)	118 (77.1)
**Portable Functional Assessment Questionnaire (Pfeiffer)**			
Severe cognitive impairment (8–10)	21 (7.42)	10 (7.7)	11 (7.2)
Moderate cognitive impairment (5–7)	73 (25.80)	29 (22.3)	44 (28.8)
Slight cognitive impairment (3–4)	48 (16.96)	26 (20.0)	22 (14.4)
Without cognitive impairment (0–2)	141 (49.82)	65 (50.0)	76 (49.7)
**The Modified Geriatric Depression Scale of Yesavage**			
Depression (10–15)	37 (13.26)	16 (12.3)	21 (14.1)
Probable depression (6–9)	154 (55.20)	65 (50.0)	89 (59.7)
No depression (0–5)	88 (31.54)	49 (37.7)	39 (26.2)

Note: we show the average values and standard deviations in brackets when the variable is numeric. We show the frequency and percentage in brackets when the variable is categorical. ***, **, and * represent statistically significant differences at 1%, 5%, and 10% between values of variables by sex.

**Table 2 ijerph-17-04009-t002:** Comparison of the mean symptom score or proportion with the symptoms in patients with and without different chronic diseases (*χ*^2^ and *t*-test).

	**Arterial Hypertension**			**Cancer**			**Cardiovascular Disease**		
	Yes	No	IC95% ^c^	Yes	No	IC95% ^c^	Yes	No	IC95% ^c^
Pain (yes) ^a^	83 (54.6)	59 (45) **	−0.21, 0.02	5 (45.5)	137 (50.4)	−0.25, 0.35	7 (46.7)	135 (50.4)	−0.22, 0.30
Anxiety ^b^	2.48	1.98 *	−1.03, 0.02	2.91	2.22	−2.06, 0.69	3.6	2.17 **	−2.72, −0.14
Constipation ^b^	1.8	1.02 **	−1.27, −0.29	2.09	1.41	−2.91, 1.54	2.53	1.37	−2.64, 0.32
Difficulty sleeping ^b^	2.81	2.29 *	−1.12, 0.08	2.73	2.56	−1.78, 1.45	3.33	2.53	−2.65, 1.03
Dyspnoea ^b^	1.03	0.61 **	−0.75, −0.10	1.64	0.81	−1.93, 0.27	1.13	0.82	−1.23, 0.61
Fatigue ^b^	2.7	2.32	−0.90, 0.15	3.45	2.49	−2.50, 0.56	3.93	2.44 *	−3.12, 0.14
Hyporexia ^b^	1.12	0.69 **	−0.79, −0.07	1.45	0.9	−1.99, 0.87	1.6	0.88	−1.93, 0.49
Nausea ^b^	0.81	0.49 **	−0.62, −0.03	1.64	0.62	−2.47, 0.44	0.87	0.65	−1.09, 0.65
Sleepiness ^b^	2.81	2.29 *	−0.71, 0.38	2.73	2.56	−0.61, 2.32	3.33	2.53	−2.97, −0.01
	**Chronic Obstructive Pulmonary Disease**					**Chronic Heart Disease**			
	Yes		No	IC95% ^c^		Yes	No		IC95% ^c^
Pain (yes) ^a^	93 (56)		49 (41.9) **	−0.26, −0.02		4 (80)	138 (49.6)		−0.66, 0.05
Anxiety ^b^	2.37		2.07	−0.84, 0.23		3	2.23		−3.79, 2.26
Constipation ^b^	1.51		1.32	−0.70, 0.33		2.2	1.42		−3.60, 2.04
Difficulty sleeping ^b^	2.69		2.4	−0.91, 0.34		5	2.53		−6.08, 1.13
Dyspnoea ^b^	0.98		0.63 **	−0.68, −0.02		2.2	0.81		−4.21, 1.44
Fatigue ^b^	2.69		2.29	−0.94, 0.15		4.4	2.49		−5.25, 1.43
Hyporexia ^b^	1.11		0.64 **	−0.83, −0.12		1.6	0.91		−2.57, 1.18
Nausea ^b^	0.76		0.52	−0.54, 0.06		2.6	0.63 *		−4.22, 0.28
Sleepiness ^b^	2.64		2.19	−1.02, 0.11		5	2.53		−5.75, 1.80
	**Chronic Liver Insufficiency**			**Chronic Renal Insufficiency**			**Dementia**		
	Yes	No	IC95% ^c^	Yes	No	IC95% ^c^	Yes	No	IC95% ^c^
Pain (yes) ^a^	13 (76.5)	129 (48.5) **	−0.49, −0.07	14 (63.6)	128 (49)	−0.36, 0.06	7 (46.7)	135 (50.4)	−0.22, 0.30
Anxiety ^b^	3.18	2.19	−2.36, 0.38	2.64	2.21	−1.43, 0.58	3.6	2.17 **	−2.72, −0.14
Constipation ^b^	2.71	1.35 **	−2.64, −0.06	1.05	1.47	−0.31, 1.16	2.53	1.37	−2.64, 0.32
Difficulty sleeping ^b^	4.06	2.47 **	−2.99, −0.18	3.14	2.52	−1.99, 0.76	3.33	2.53	−2.65, 1.03
Dyspnoea ^b^	1.47	0.80 **	−1.27, −0.08	1.14	0.81	−1.16, 0.51	1.13	0.82	−1.23, 0.61
Fatigue ^b^	4.18	2.42 **	−2.82, −0.70	2.77	2.5	−1.28, 0.74	3.93	2.44 *	−3.12, 0.14
Hyporexia ^b^	2.47	0.82 **	−2.72, −0.58	0.95	0.92	−0.59, 0.51	1.6	0.88	−1.93, 0.49
Nausea ^b^	2.24	0.56 **	−2.87, −0.48	1	0.63	−1.05, 0.32	0.87	0.65	−1.09, 0.65
Sleepiness ^b^	4.06	2.47 **	−2.64, 0.10	3.14	2.52	−1.20, 0.71	3.33	2.53	−2.97, −0.01
	**Diabetes**			**Joint Diseases**			**Pressure Ulcer**		
	Yes	No	IC95% ^c^	Yes	No	IC95% ^c^	Yes	No	IC95% ^c^
Pain (yes) ^a^	51 (63.7)	91 (44.8) **	−0.32, −0.06	15 (83.3)	127 (47.9) **	−0.54, −0.17	8 (47.1)	134 (50.4)	−0.21, 0.28
Anxiety ^b^	2.61	2.10 *	−1.09, 0.07	3.72	2.15 **	−2.74, −0.41	3.82	2.15 **	−3.11, −0.25
Constipation ^b^	2.16	1.15 ***	−1.60, −0.43	3.22	1.31 **	−3.26, −0.56	0.88	1.47	−0.15, 1.33
Difficulty sleeping ^b^	3.23	2.31 **	−1.63, −0.20	4.22	2.46 **	−3.29, −0.24	3.94	2.48	−3.31, 0.39
Dyspnoea ^b^	1.21	0.69 **	−0.90, −0.14	1.72	0.78 **	−1.70, −0.19	1.76	0.78 *	−2.17, 0.19
Fatigue ^b^	3.21	2.25 **	−1.54, −0.38	3.72	2.44 **	−2.48, −0.08	2.88	2.5	−1.61, 0.85
Hyporexia ^b^	1.39	0.73 **	−1.07, −0.23	2.56	0.81 ***	−2.60, −0.89	1.06	0.91	−0.99, 0.69
Nausea ^b^	1.21	0.44 ***	−1.19, −0.35	2.39	0.54 **	−2.92, −0.77	0.41	0.68	−0.20, 0.73
Sleepiness ^b^	3.23	2.31 **	−1.24, −0.00	4.22	2.46 **	−2.31, 0.56	2.94	2.42	−1.94, 0.91

Note: ^a^ we show the frequency and percentage in brackets when the variable is categorical and ^b^ the average when the variable is numeric. ***, **, and * represent statistically significant differences at 1%, 5%, and 10% between values of variables with and without chronic disease. ^c^ IC95%: 95% confidence interval for the difference (No-Yes) between the mean symptom score (or proportion with the symptom) among those without the disease (No) and among those with the disease (Yes).

**Table 3 ijerph-17-04009-t003:** Comparative analysis of functional characteristics of patients with different numbers of chronic diseases.

	Number of Chronic Diseases ^a^			Chi-Squared Test^b^ (*p*-Value) or Kruskal–Wallis test ^c^ (*p*-Value)
	None or 1 Chronic Disease	2 Chronic Diseases	3 or More Chronic Diseases	
**Barthel index**				**38.438 (0.000) ^b^**
Severe dependence	10 (7.7)	9 (10.3)	12 (18.2)	
Moderate dependence	5 (3.8)	12 (13.8)	14 (21.2)	
Slight dependence	55 (42.3)	38 (43.7)	35 (53.0)	
Independence	60 (46.2)	28 (32.2)	5 (7.6)	
**Detection of Emotional Distress (DED) scale**				
With emotional distress	53 (42.7)	40 (48.8)	30 (48.4)	0.926 (0.629) ^b^
Without emotional distress	71 (57.3)	42 (51.2)	32 (51.6)	
**Katz Index**				**12.991 (0.011) ^b^**
Independence	63 (23.1)	25 (28.7)	25 (37.9)	
Moderate dependence	37 (28.5)	39 (44.8)	31 (47.0)	
Severe Dependence	30 (48.5)	23 (26.4)	10 (15.2)	
**Karnofsky Performance Status (KPS)** (100 = normal to 0 = dead)	85.00 (15.36)	76.21 (20.76)	66.82 (23.35)	**31.555 (0.000) ^c^**
**Lawton and Brody Instrumental Activities of Daily Living (IADL) Scale**				
Dependence	16 (12.3)	17 (19.5)	11 (16.9)	3.334 (0.504) ^b^
Semi-dependence	80 (61.5)	45 (51.7)	34 (52.3)	
Independence	34 (26.2)	25 (28.7)	20 (30.8)	
**The Modified Mini-Mental State Test**				
Cognitive deficit	29 (22.3)	17 (19.5)	21 (31.8)	3.379 (0.185) ^b^
Without cognitive deficit	101 (77.7)	70 (80.5)	45 (68.2)	
**Portable Functional Assessment Questionnaire (Pfeiffer)**				
Severe cognitive impairment	10 (7.7)	8 (9.2)	3 (4.5)	**14.802 (0.021) ^b^**
Moderate cognitive impairment	30 (10.8)	19 (21.8)	15 (22.7)	
Slight cognitive impairment	14 (23.1)	19 (21.8)	24 (36.4)	
Without cognitive impairment	76 (58.5)	41 (47.1)	24 (36.4)	
**The Modified Geriatric Depression Scale of Yesavage**				
Depression	12 (9.2)	15 (17.6)	10 (15.6)	10.204 (0.037) ^b^
Probable depression	83 (26.9)	26 (30.6)	27 (42.2)	
No depression	35 (63.8)	44 (51.8)	27 (42.2)	

Note: ^a^ chronic diseases: cancer, dementia, diabetes, cardiovascular disease, arterial hypertension, joint diseases, chronic obstructive pulmonary disease, chronic heart disease, chronic liver insufficiency, chronic renal insufficiency, pressure ulcer. We show the average values and standard deviations in brackets when a Kruskal–Wallis test ^c^ was applied or frequency and percentages in brackets when a Chi-squared test ^b^ was applied

**Table 4 ijerph-17-04009-t004:** Associations between dependence level (Barthel Index) and different chronic diseases.

Outcome Variable: Barthel Index (Dependence)	OR_CRUDE_ (IC95%)	OR_ADJUSTED_ (IC95%)
**Number of chronic diseases ^a^** (None or one chronic disease)		
Two chronic diseases	**2.44 (1.18, 5.13)** **	
Three or more chronic diseases	**4.98 (2.43, 10.55)** ***	
**Visual disorders** (yes)	**2.03 (1.14, 3.60)** **	1.63 (0.80, 3.34)
**Cancer** (yes)	**4.63 (1.35, 16.59)** **	**9.23 (2.24, 43.03)** **
**Dementia** (yes)	**4.53 (1.56, 13.45)** **	**4.84 (1.43, 17.06)** **
**Diabetes** (yes)	1.70 (0.93, 3.07)	1.28 (0.63, 2.56)
**Pain** (yes)	0.72 (0.41,1.27)	1.03 (0.51, 2.10)
**Cardiovascular disease** (yes)	2.42 (0.31, 14.94)	1.12 (0.09, 9.61)
**Joint diseases** (yes)	1.87 (0.63, 5.03)	0.88 (0.23, 3.03)
**Chronic Obstructive Pulmonary Disease** (yes)	1.26 (0.71, 2.27)	1.12 (0.58, 2.18)
**Arterial Hypertension** (yes)	**2.56 (1.41, 4.80)** **	**2.68 (1.32, 5.70)** **
**Chronic Liver Insufficiency** (yes)	2.05 (0.68, 5.62)	1.22 (0.36, 3.88)
**Chronic Renal Insufficiency** (yes)	1.75 (0.64, 4.36)	1.55 (0.46, 4.76)
**Pressure ulcer** (yes)	0.75 (0.17, 2.40)	0.28 (0.03, 1.31)
**Age (centered on 76 years)**	1.03 (0.9, 1.07)	1.02 (0.98, 1.06)
**Sex** (male)	0.81 (0.46, 1.43)	0.77 (0.39, 1.48)
**BMI** (Normal weight)		
Underweight	1.57 (0.51, 4.48)	2.53 (0.73, 8.26)
Overweight	0.77 (0.39, 1.48)	0.93 (0.45, 1.92)
Obesity	**0.36 (0.12, 0.92)** **	**0.25 (0.07, 0.74)** **

Note: the Barthel index variable was re-coded as a dichotomous variable (coded as “Dependence” those who had “severe and moderate dependence” and as “Independence” those who had “slight dependence” or were independent). ***, **, and * represent statistically significant differences at 1%, 5%, and 10%. ^a^ Chronic diseases: cancer, dementia, diabetes, cardiovascular disease, arterial hypertension, joint diseases, chronic obstructive pulmonary disease, chronic heart disease, chronic liver insufficiency, chronic renal insufficiency, pressure ulcer.

**Table 5 ijerph-17-04009-t005:** Associations between clinical and functional situation (KPS) and different chronic diseases or the number of chronic diseases ^a^.

Outcome Variable: Karnofsky Performance Status (KPS) (<85 Points)	OR_CRUDE_ (IC95%)	OR_ADJUSTED_ (IC95%)
**Number of chronic diseases** (None or one chronic disease)		
Two chronic diseases	**1.99 (1.15, 3.48)** **	
Three or more chronic diseases	**3.24 (1.74, 6.18)** ***	
**Visual disorders** (yes)	**2.33 (1.42, 3.88)** **	1.75 (0.93, 3.29)
**Cancer** (yes)	**9.29 (1.74, 171.68)** **	13,831,989.33 (0.00, …) ^b^
**Dementia** (yes)	**13.39 (02.64, 244.31)** **	**9.94 (1.80, 186.55)** **
**Diabetes** (yes)	1.68 (0.99, 2.88)	1.31 (0.69, 2.47)
**Pain** (yes)	0.74 (0.46, 1.19)	0.97 (0.54, 1.77)
**Joint diseases** (yes)	1.40 (0.54, 3.92)	1.07 (0.32, 3.76)
**Chronic Obstructive Pulmonary Disease** (yes)	1.37 (0.86, 2.21)	1.35 (0.77, 2.39)
**Arterial Hypertension** (yes)	**1.67 (1.04, 2.68)** **	1.67 (0.93, 3.00)
**Chronic Liver Insufficiency** (yes)	1.27 (0.47, 3.58)	0.55 (0.17, 1.87)
**Chronic Renal Insufficiency** (yes)	1.97 (0.80, 5.30)	1.49 (0.50, 4.71)
**Pressure ulcer** (yes)	1.65 (0.61, 4.91)	1.27 (0.40, 4.27)
**Age (centered on 76 years)**	**1.07 (1.04, 1.11)** ***	**1.07 (1.04, 1.12)** ***
**Sex** (male)	**0.57 (0.36, 0.92)** **	**0.53 (0.30, 0.91)** **
**BMI** (Normal weight)		
Underweight	2.51 (0.83, 9.29)	**3.38 (1.01, 13.75) ***
Overweight	0.83 (0.47, 1.44)	0.96 (0.51, 1.79)
Obesity	0.68 (0.34, 1.34)	0.59 (0.26, 1.30)

Note: ***, **, and * represent statistically significant differences at 1%, 5%, and 10%. ^a^ chronic diseases: cancer, dementia, diabetes, cardiovascular disease, arterial hypertension, joint diseases, chronic obstructive pulmonary disease, chronic heart disease, chronic liver insufficiency, chronic renal insufficiency, pressure ulcer. ^b^ “…”: value greater than 0 is very high. KPS: the higher the score, the better the functional and clinical situation.

**Table 6 ijerph-17-04009-t006:** Associations between depression (the modified geriatric depression scale of Yesavage) and different chronic conditions.

Outcome Variable: The Modified Geriatric Depression Scale of Yesavage (Depression or Probable Depression)	OR_CRUDE_ (IC95%)	OR_ADJUSTED_ (IC95%)
**Number of chronic diseases ^a^** (None or one chronic disease)		
Two chronic diseases	**1.65 (0.94, 2.88) ***	
Three or more chronic diseases	**2.42 (1.32, 4.50)** **	
**Visual disorders** (yes)	**2.10 (1.28, 3.46)** **	1.23 (0.62, 2.41)
**Cancer** (yes)	**5.90 (1.48, 39.16)** **	**8.29 (1.30, 165.28)** *
**Dementia** (yes)	1.08 (0.37, 3.10)	0.42 (0.11, 1.54)
**Diabetes** (yes)	**1.99 (1.17, 3.41)** **	**2.03 (1.03, 4.05)** **
**Pain** (yes)	**0.62 (0.38, 0.99)** **	0.63 (0.33, 1.17)
**Cardiovascular disease** (yes)	5.06 (0.74, 99.63)	4.03 (0.41, 93.29)
**Joint diseases** (yes)	1.59 (0.61, 4.28)	0.89 (0.28, 2.80)
**Chronic Obstructive Pulmonary Disease** (yes)	1.27 (0.79, 2.06)	1.19 (0.66, 2.13)
**Arterial Hypertension** (yes)	1.39 (0.87, 2.25)	1.14 (0.62 2.11)
**Chronic Liver Insufficiency** (yes)	2.14 (0.77, 6.47)	1.82 (0.57, 6.27)
**Chronic Renal Insufficiency** (yes)	1.03 (0.42, 2.47)	0.60 (0.19, 1.81)
**Pressure ulcer** (yes)	**2.38 (0.88, 7.09) ***	2.56 (0.74, 9.46)
**Age (centered on 76 years)**	1.01 (0.98, 1.04)	0.99 (0.95, 1.03)
**Sex** (male)	0.67 (0.42, 1.08)	0.73 (0.40, 1.32)
**Civil status** (Married)		
Divorced	0.80 (0.16, 3.39)	0.88 (0.13, 5.08)
Unmarried/Single	0.96 (0.52, 1.77)	1.59 (0.71, 6.57)
Widowed	1.36 (0.78, 2.38)	1.06 (0.52, 2.14)
**Gijon’s Social-familial evaluation scale** (Normal social situation)		
Social risk situation	**3.81 (2.12, 6.96)** ***	**2.46 (1.21, 5.06)** **
Social problem	1.30 (0.71, 2.39)	1.25 (0.58, 2.70)
**Barthel index** (Dependence)	**2.09 (1.18, 3.75)** **	1.16 (0.54, 2.51)
**Katz Index** (Independence)		
Moderate dependence	2.03 (1.18, 3.52)	1.24 (0.61, 2.54)
Severe Dependence	1.51 (0.81, 2.85)	1.45 (0.66, 3.18)
**Karnofsky Performance Status (KPS) (<85 points)**	**4.26 (2.58, 7.14)** ***	**2.84 (1.48, 5.54)** **
**Lawton and Brody Instrumental Activities of Daily Living (IADL) Scale** (Independence)		
Semi-dependence	**2.07 (1.18, 3.69)** **	1.73 (0.86, 3.55)
Dependence	**2.22 (1.04, 4.81)** **	1.88 (0.72, 4.92)

Note: ***, **, and * represent statistically significant differences at 1%, 5%, and 10%. Dependent variable: The Modified Geriatric Depression Scale coded as a dichotomous variable (coded as “depression” those who were classified as “depression” or “probable depression”, and as “no depression” those who were classified as “no depression”). KPS: the higher the score, the better the functional and clinical situation. ^a^ Chronic diseases: cancer, dementia, diabetes, cardiovascular disease, arterial hypertension, joint diseases, chronic obstructive pulmonary disease, chronic heart disease, chronic liver insufficiency, chronic renal insufficiency, pressure ulcer.

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
