# Peer review of "Chronic Diseases and Associated Factors among Older Adults in Loja, Ecuador"

_ijerph, 2020, doi:10.3390/ijerph17114009_

Round 1

Reviewer 1 Report

The authors have an interesting data set, however, there are serious deficiencies in formulation of the study objectives, the sample selection method, and the data analysis and presentation. I advise that the manuscript is not suitable for publication. However, if the authors wish to revise their analysis and manuscript, and re-submit to this or another journal, then I make the following recommendations.

  1. The title is not consistent with the stated objectives, or the study sample. You can't measure prevalence unless you have a random sample of the population. I suggest changing it to something like, "Health-related quality of life among elderly people with chronic diseases in Loja, Ecuador: description and associated markers."

  1. What was the target population (who were the study subjects intended to represent)? How were the subjects selected and recruited? What was the participation rate? Show evidence that the study subjects were representative (age, gender, geographic location, socioeconomic status, etc., of the target population).

  1. The description of sample size calculation is not informative, and contains obvious contradictions. Did you use Epi Info software or the GRANMO tool? Were you trying to calculate sample size for a survey (i.e., meet a desired level of precision) or calculate group sizes for a comparison of two groups (i.e., have desired amount of power to detect a meaningful difference between the groups)?

  1. Table 1, characteristics of subjects

The clinimetric characteristics are measured on numeric scales, however, you have grouped the scores into categories (e.g., no, slight, moderate, severe). That is not wrong, but you may be losing statistical power. However, if you think the categories are more understandable to readers, then I recommend also stating the numeric ranges each category represents, and stating the mean and s.d. scores of groups (or medians in cases where the scores are ordinal-scale, or not normally distributed and you intend to compare groups using non-parametric tests).

  1. Table 2, comparison of means

This compares means of scores on subscales of the Edmonton Symptom Assessment Scale (ESAS), between those with and without various chronic diseases.

For most of the diseases, the number of subjects with the disease is so small that you have little power to detect meaningful differences in symptom scores between those with and those without the disease. In such cases, p-value greater than 0.05 is not statistically significant, but the difference still could be of practical importance, you just didn't have a large enough sample size to show that it was unlikely to have been caused by chance. I recommend showing the 95% confidence limits for the difference, instead of p-values.

  1. Table 3, comparative analysis of clinimetric characteristics

It is very difficult to see the clinimetric differences between the "number of chronic disease" groups. I recommend showing the column percentages (all the percentages in a column sum to 100%) instead of the row percentages. Also, for each clinimetric variable I recommend showing the median score in each "number of chronic disease" group. Test the null hypothesis that the three medians are the same. Put the name of the clinimetric variable at the top of the left-most column, not in the middle of the row, on top of the number of disease category columns.

  1. Table 4: Associations between dependence level (Barthel Index) and different chronic diseases.

This table and the analysis it presumes to describe are incomprehensible. In logistic regression, the outcome variable must be dichotomous (coded as 1=yes, 0=no). If you have coded Barthel Index (a numeric score) into 4 categories (total dependence, moderate dependence, slight dependence, independence) then I don't see how you could use it as the outcome variable. The explanation in the text that the odds ratio represents risk of developing total dependence based on reported level of dependence (moderate, slight, or independence) is bizarre. I recommend re-coding Barthel Index as a dichotomous variable, then perform logistic regression analysis.

  1. Table 4: Associations between dependence level (Barthel Index) and number of chronic diseases.

You call this Table 4.1 in the text.

This table and the analysis it presumes to describe are incomprehensible. In logistic regression, the outcome variable must be dichotomous (coded as 1=yes, 0=no). If you have coded Barthel Index into 4 categories (total dependence, moderate dependence, slight dependence, independence) then I don't see how you could use it as the outcome variable. The explanation in the text that the odds ratio represents risk of developing total dependence based on reported level of dependence (moderate, slight, or independence) is bizarre. I recommend re-coding Barthel Index as a dichotomous variable, then perform logistic regression analysis.

  1. Table 5. Associations between clinical and functional situation (KPS) and different chronic diseases or the number of chronic diseases.

This appears to report results of linear regression analysis with Karnofsky Performance Status (KPS) as the outcome variable. This is not an appropriate method of analysis for KPS because KPS numeric score (0=dead, 10=moribund, 20=very sick, hospitalization necessary, .... 90=minor signs or symptoms but able to carry on normal activity, 100=normal with no complaints) is not an interval-ratio scale variable, it is an ordinal scale variable. If it were interval-ratio scale, then a change of say, -10 would have the same magnitude of effect anywhere along the scale. Obviously that is not the case, because a decline from 20 to 10 has a very different meaning than a decline from 100 to 90. The estimated regression coefficients are therefore impossible to interpret. I recommend re-coding KPS as a dichotomous variable (85 to 100 coded as "1" and <85 coded as "0") and then performing logistic regression analysis.

  1. Table 6. Associations between depression (The Modified Geriatric Depression Scale of Yesavage) and different chronic conditions.

This table, the analysis it presumes to describe, and the explanation in the text are incomprehensible. In logistic regression, the outcome variable must be dichotomous (coded as 1=yes, 0=no). If you have coded the Modified Geriatric Depression Scale (a numeric score) into 3 categories (depression, probable depression, no depression) then I don't see how you could use it as the outcome variable. I recommend re-coding the Modified Geriatric Depression Scale as a dichotomous variable, then perform logistic regression analysis.

  1. Table 7. Associations between the number of chronic conditions and self-reported social risk with depression (The Modified Geriatric Depression Scale of Yesavage).

This table, the analysis it presumes to describe, and the explanation in the text are incomprehensible. In logistic regression, the outcome variable must be dichotomous (coded as 1=yes, 0=no). If you have coded the Modified Geriatric Depression Scale into 3 categories (depression, probable depression, no depression) then I don't see how you could use it as the outcome variable. I recommend re-coding the Modified Geriatric Depression Scale as a dichotomous variable, then perform logistic regression analysis.

Author Response

The authors have an interesting data set, however, there are serious deficiencies in formulation of the study objectives, the sample selection method, and the data analysis and presentation. I advise that the manuscript is not suitable for publication. However, if the authors wish to revise their analysis and manuscript, and re-submit to this or another journal, then I make the following recommendations.

Dear reviewer, thanks for your observations that help us to improve our manuscript. We will carefully take into account all your suggestions.

  1. The title is not consistent with the stated objectives, or the study sample. You can't measure prevalence unless you have a random sample of the population. I suggest changing it to something like, "Health-related quality of life among elderly people with chronic diseases in Loja, Ecuador: description and associated markers."

Thank you for this recommendation that sadly we cannot attend because we consider that to include “health-related quality of life” is not suitable because we did not measure this aspect. Nevertheless, We agree with your comment that we should change the title and delete the word “Prevalence”. The new title chosen is Chronic diseases among elderly people in Loja, Ecuador: description and associated factors.

  1. What was the target population (who were the study subjects intended to represent)? How were the subjects selected and recruited? What was the participation rate? Show evidence that the study subjects were representative (age, gender, geographic location, socioeconomic status, etc., of the target population).

Regarding this question, we recruited patients with, at least, one chronic disease or a risk factor to develop another chronic condition, such as obesity to know what is patients´ daily life situation. Patients were recruited from health centers and geriatric institutions associated with that primary health centers, in Loja, Ecuador. All patients that met inclusion criteria were eligible to be included in the study, but, in order to accomplish the randomization, a simple random sampling was carried out to recruit patients. (Please, see section “Materials and Methods”, lines 102-108).

On the other hand, as we show in table 1, sociodemographic characteristics are similar to the population living in the area. If we consult national statistics in Ecuador, we observe that are similar https://www.ecuadorencifras.gob.ec/wp-content/descargas/Manu-lateral/Resultados-provinciales/loja.pdf. However, we observe that our sample is formed by more men than women, when the situation is just the reverse (49,2% men vs 50,8%). We consider that this may be due to men decided to participate more than women (54% vs 46%).

Sample was selected by chronic process according to the main chronic diseases and data published about Loja on this website  (reference 8 in the paper) https://public.tableau.com/views/cronicas_2014_0/ANUARIO?:embed=y&:showVizHome=no&:display_count=y&:display_static_image=y

  1. The description of sample size calculation is not informative, and contains obvious contradictions. Did you use Epi Info software or the GRANMO tool? Were you trying to calculate sample size for a survey (i.e., meet a desired level of precision) or calculate group sizes for a comparison of two groups (i.e., have desired amount of power to detect a meaningful difference between the groups)?

Thank you for this comment. We have deleted the following sentence due to a mistake: “The sample size for this study was calculated using Epi Info software”. We used the GRANMO tool. In relation to calculation of sample size, we calculated the sample size for a questionnaire.

  1. Table 1, characteristics of subjects

The clinimetric characteristics are measured on numeric scales, however, you have grouped the scores into categories (e.g., no, slight, moderate, severe). That is not wrong, but you may be losing statistical power. However, if you think the categories are more understandable to readers, then I recommend also stating the numeric ranges each category represents, and stating the mean and s.d. scores of groups (or medians in cases where the scores are ordinal-scale, or not normally distributed and you intend to compare groups using non-parametric tests).

Thank you for this observation. As you mention, we tried to show data more understandable to readers, so we grouped into categories. 

  1. Table 2, comparison of means

This compares means of scores on subscales of the Edmonton Symptom Assessment Scale (ESAS), between those with and without various chronic diseases.

For most of the diseases, the number of subjects with the disease is so small that you have little power to detect meaningful differences in symptom scores between those with and those without the disease. In such cases, p-value greater than 0.05 is not statistically significant, but the difference still could be of practical importance, you just didn't have a large enough sample size to show that it was unlikely to have been caused by chance. I recommend showing the 95% confidence limits for the difference, instead of p-values.

Thank you very much for this comment. We agree with you and we have incorporated the 95% confidence intervals for the difference instead of p-values. 

  1. Table 3, comparative analysis of clinimetric characteristics

It is very difficult to see the clinimetric differences between the "number of chronic disease" groups. I recommend showing the column percentages (all the percentages in a column sum to 100%) instead of the row percentages. Also, for each clinimetric variable I recommend showing the median score in each "number of chronic disease" group. Test the null hypothesis that the three medians are the same. Put the name of the clinimetric variable at the top of the left-most column, not in the middle of the row, on top of the number of disease category columns.

Thank you for this comment. We have incorporated your suggestions to the table 3, adding the column percentages and putting the name of the clinimetric variable at the top of the left-most column.

  1. Table 4: Associations between dependence level (Barthel Index) and different chronic diseases.

This table and the analysis it presumes to describe are incomprehensible. In logistic regression, the outcome variable must be dichotomous (coded as 1=yes, 0=no). If you have coded Barthel Index (a numeric score) into 4 categories (total dependence, moderate dependence, slight dependence, independence) then I don't see how you could use it as the outcome variable. The explanation in the text that the odds ratio represents risk of developing total dependence based on reported level of dependence (moderate, slight, or independence) is bizarre. I recommend re-coding Barthel Index as a dichotomous variable, then perform logistic regression analysis.

Thank you for this comment. In accordance with the literature (we attach bibliography related to this type of models), the logistic regression analysis can be used with categorical variables with more than 2 response categories. In this case, the logistic regression analysis is known as multinomial logistic regression analysis instead of bivariate or binomial logistic regression model. We agree with you about it is more difficult understand its findings. Then, we have re-coded Barthel Index as a dichotomous variable (coded as “Dependence” those who had “severe and moderate dependence” and as “Independence” those who had “slight dependence” or were independents), then we have performed logistic regression analysis. Then, we have modified the table 4 completely and the results related to this.

Härdle, W. K., & Simar, L. (2015). Applied multivariate statistical analysis, fourth edition. Applied Multivariate Statistical Analysis, Fourth Edition. https://doi.org/10.1007/978-3-662-45171-7

Kuonen, D. (2004). Book Review: Regression modeling strategies: with applications to linear models, logistic regression, and survival analysis. Statistical Methods in Medical Research (Vol. 13). https://doi.org/10.1177/096228020401300512

Jokela, M., García-Velázquez, R., Airaksinen, J., Gluschkoff, K., Kivimäki, M., & Rosenström, T. (2019). Chronic diseases and social risk factors in relation to specific symptoms of depression: Evidence from the U.S. national health and nutrition examination surveys. Journal of Affective Disorders, 251(March), 242–247. https://doi.org/10.1016/j.jad.2019.03.074

Van Calster, B., Hoorde, K. Van, Vergouwe, Y., Bobdiwala, S., Condous, G., Kirk, E., … Steyerberg, E. W. (n.d.). Validation and updating of risk models based on multinomial logistic regression. https://doi.org/10.1186/s41512-016-0002-x

Shin, S. M. (2017). Prevalence and trends of pain associated with chronic diseases and personal out-of-pocket medical expenditures in Korea. Korean J Pain, 30(2). https://doi.org/10.3344/kjp.2017.30.2.142

  1. Table 4: Associations between dependence level (Barthel Index) and number of chronic diseases.

You call this Table 4.1 in the text.

This table and the analysis it presumes to describe are incomprehensible. In logistic regression, the outcome variable must be dichotomous (coded as 1=yes, 0=no). If you have coded Barthel Index into 4 categories (total dependence, moderate dependence, slight dependence, independence) then I don't see how you could use it as the outcome variable. The explanation in the text that the odds ratio represents risk of developing total dependence based on reported level of dependence (moderate, slight, or independence) is bizarre. I recommend re-coding Barthel Index as a dichotomous variable, then perform logistic regression analysis.

Thank you for this comment. We have re-coded Barthel Index as a dichotomous variable, then we have performed logistic regression analysis. Then, we have added the findings in the table 4 and have modified the results related to this. 

  1. Table 5. Associations between clinical and functional situation (KPS) and different chronic diseases or the number of chronic diseases.

This appears to report results of linear regression analysis with Karnofsky Performance Status (KPS) as the outcome variable. This is not an appropriate method of analysis for KPS because KPS numeric score (0=dead, 10=moribund, 20=very sick, hospitalization necessary, .... 90=minor signs or symptoms but able to carry on normal activity, 100=normal with no complaints) is not an interval-ratio scale variable, it is an ordinal scale variable. If it were interval-ratio scale, then a change of say, -10 would have the same magnitude of effect anywhere along the scale. Obviously that is not the case, because a decline from 20 to 10 has a very different meaning than a decline from 100 to 90. The estimated regression coefficients are therefore impossible to interpret. I recommend re-coding KPS as a dichotomous variable (85 to 100 coded as "1" and <85 coded as "0") and then performing logistic regression analysis.

Thank you very much for this comment. We have re-coded KPS as a dichotomous variable according to your recommendations and have performed a logistic regression model. Then, we have modified the table 5 completely and the results related to this. In addition, this analysis is detailed in section “Material and methods” (lines 147-149): “The associations between functional and clinical situations measured through KPS Index and different chronic diseases were studied using bivariate logistic regression analysis, after re-coding the KPS Index as a dichotomous variable.”

  1. Table 6. Associations between depression (The Modified Geriatric Depression Scale of Yesavage) and different chronic conditions.

This table, the analysis it presumes to describe, and the explanation in the text are incomprehensible. In logistic regression, the outcome variable must be dichotomous (coded as 1=yes, 0=no). If you have coded the Modified Geriatric Depression Scale (a numeric score) into 3 categories (depression, probable depression, no depression) then I don't see how you could use it as the outcome variable. I recommend re-coding the Modified Geriatric Depression Scale as a dichotomous variable, then perform logistic regression analysis.

Thank you for this comment. In accordance with the literature (we attach bibliography related to this type of models), the logistic regression analysis can be used with categorical variables with more than 2 response categories. In this case, the logistic regression analysis is known as multinomial logistic regression analysis instead of bivariate or binomial logistic regression model. We agree with you about it is more difficult understand its findings, but in this case, we think that re-coded this variable as a dichotomous variable could loss statistical power. In addition, we cannot assume those who have “probable depression” as “depression” due to the very different frequencies of those categories in this variable.

Härdle, W. K., & Simar, L. (2015). Applied multivariate statistical analysis, fourth edition. Applied Multivariate Statistical Analysis, Fourth Edition. https://doi.org/10.1007/978-3-662-45171-7

Kuonen, D. (2004). Book Review: Regression modeling strategies: with applications to linear models, logistic regression, and survival analysis. Statistical Methods in Medical Research (Vol. 13). https://doi.org/10.1177/096228020401300512

Jokela, M., García-Velázquez, R., Airaksinen, J., Gluschkoff, K., Kivimäki, M., & Rosenström, T. (2019). Chronic diseases and social risk factors in relation to specific symptoms of depression: Evidence from the U.S. national health and nutrition examination surveys. Journal of Affective Disorders, 251(March), 242–247. https://doi.org/10.1016/j.jad.2019.03.074

Van Calster, B., Hoorde, K. Van, Vergouwe, Y., Bobdiwala, S., Condous, G., Kirk, E., … Steyerberg, E. W. (n.d.). Validation and updating of risk models based on multinomial logistic regression. https://doi.org/10.1186/s41512-016-0002-x

Shin, S. M. (2017). Prevalence and trends of pain associated with chronic diseases and personal out-of-pocket medical expenditures in Korea. Korean J Pain, 30(2). https://doi.org/10.3344/kjp.2017.30.2.142

  1. Table 7. Associations between the number of chronic conditions and self-reported social risk with depression (The Modified Geriatric Depression Scale of Yesavage).

This table, the analysis it presumes to describe, and the explanation in the text are incomprehensible. In logistic regression, the outcome variable must be dichotomous (coded as 1=yes, 0=no). If you have coded the Modified Geriatric Depression Scale into 3 categories (depression, probable depression, no depression) then I don't see how you could use it as the outcome variable. I recommend re-coding the Modified Geriatric Depression Scale as a dichotomous variable, then perform logistic regression analysis.

Thank you for this comment. We have discussed among research team this suggestion because we disagree with the idea to dichotomize the variable. If we do this, we are losing the category “probable depression” that encourage us to follow this kind of patients. It is interesting to have this category because if we remove it, relevant information would be missing. Many patients were categorized with this label.

Reviewer 2 Report

  • Title: Prevalence and factors related to chronic diseases in Loja, Ecuador

Beware you do not address all chronic diseases in your study. It is a synthetic title which must be more precise (target population, date, etc...).

  • The introduction is well conducted and argued but at no time is it discussed what will be one of the specific objectives of the study i.e. the consequences in terms of signs and symptoms, the existing associations between the level of dependence, the clinical level and the functional level, etc....

There may be too many secondary objectives. No need to specify at this stage the indices used

  • Why is the study taking place over 1 year for such a small number of subjects included (N=247)?

  • We used numerous recruitment strategies, with random selection of participants following a 1:1 selection ratio. Individuals were interviewed by trained staff who had no medical relationship with the patients. These patients were attended to in healthcare centers of the Health Ministry of Health, Ecuador.

What does "1:1 selection ratio" mean for an observational study?

  • The study included patients in the 60 years and over age group.

            Why hasn't an upper limit been defined?

  • These patients were attended to in healthcare centers of the Health Ministry of Health, Ecuador.

One of the major problems of elderly populations is the geographical access to health centers, due to loss of autonomy, difficult access to transport, financial difficulties, etc.... Not to mention those living in institutions: Your sampling is biased.

  • Did you stratify by age group and gender?

  • Your inclusion-exclusion criteria are very limited

  • You do not follow the STROBE guidelines process for writing and reading observational studies

  • You write: "We used numerous recruitment strategies".

What does that mean?

  • Sample size

You will refer to diabetes. Why, when your research is multi-morbidity. In the calculation of your sample size, must intervene the range of age of your patients, sex, factors studied, SES, location, etc... The reasoning is based on a multistratified pathinder cluster approach as recommended by the WHO.

It is difficult to meet your specific objectives on such a small and unrepresentative base of subjects from the general population.

  • Ethical considerations

Please add the agreement number of the department

  • You wrote: "Written consent was required for all participants".

In this age group, it is impossible to have written consent for the entire population: Subjects who are no  autonomous, subjects under guardianship, dependent subjects, Alzheimer, etc... Please can you specify?

  • 1. Demographic Characteristics.

There are variables that are irrelevant or ethically questionable. Ex: Marital status Do you want to talk about one of the criteria that is part of the studies of seniors returning to isolation? If yes, there are 3-4 additional criteria to specify. If not, no interest.

  • Religion

Can you explain the logic of including this variable???? It is ethically questionable.

  • The statistical analysis is well done.
  • Can you explain why you are talking about the average number of pathologies vs. Qualitative (ex: Number of chronic diseases)
  • "As a limitation we affirm these data are from only one region of Ecuador and may not be representative of the whole country."

Are the data representative of Loja?

  • It is necessary to develop strategies to improve care provided to chronic patients in Ecuador in order to reduce associated morbidity.

Please can you in the discussion specify some tracks or recommendations made on this subject

  • To decrease geriatric depression associated with chronicity, it is recommended to improve autonomy in daily life activities.

Unless I'm mistaken, this is the first time you've used the word "geriatric" in its generic sense. All other terms are related to "The Modified Geriatric Depression Scale of Yesavage". This brings me back to the comments about the profile of your study population.

  • "Targeted interventions to reduce the morbidity of chronic diseases in Ecuador are essential to improve health-related quality of life."

This term is used 4 times in this paper, 1 in summary and 2 in discussion. How does it relate to your results? Your introduction and discussion talk about GBD that are not quality of life.

  • You wrote: "At present, it has become evident that measuring years lived with disability (YLDs) is important for estimating the burden of non-lethal diseases and, consequently, for planning health service".

Yes, but concretely, how do you translate in your results the morbidity results observed in YLDS?

Author Response

  • Title: Prevalence and factors related to chronic diseases in Loja, Ecuador

Beware you do not address all chronic diseases in your study. It is a synthetic title which must be more precise (target population, date, etc...).

Thank you for this observation, in fact, we changed the title and now is:  Chronic diseases among elderly people in Loja, Ecuador: description and associated factors.

  • The introduction is well conducted and argued but at no time is it discussed what will be one of the specific objectives of the study i.e. the consequences in terms of signs and symptoms, the existing associations between the level of dependence, the clinical level and the functional level, etc....

There may be too many secondary objectives. No need to specify at this stage the indices used

Thank you for this comment. We deleted these aspects following your suggestion. 

  • Why is the study taking place over 1 year for such a small number of subjects included (N=247)?

Thank you for this question. A year was the timing required to recruit and assess all participants but is a cross-sectional epidemiological research (Please, see lines 78-79).

  • We used numerous recruitment strategies, with random selection of participants following a 1:1 selection ratio. Individuals were interviewed by trained staff who had no medical relationship with the patients. These patients were attended to in healthcare centers of the Health Ministry of Health, Ecuador.

What does "1:1 selection ratio" mean for an observational study?

We appreciate your question. We recruited patients following a random sampling technique to guarantee representability. To clarify this aspect, the sentence has been changed to:  

Patients were selected by using simple random sampling technique” (Please, see line 105).

  • The study included patients in the 60 years and over age group.

            Why hasn't an upper limit been defined?

Thank you for your question. We stablished to include patients in the 60 years and over because WHO determines older population at this age. At the moment, there is no United Nations standard numerical criterion, but the UN agreed cutoff is 60+ years to refer to the older population. Although there are commonly used definitions of old age, there is no general agreement on the age at which a person becomes old. This statement has been included in lines 91-93.

(https://www.who.int/healthinfo/survey/ageingdefnolder/en/)

  • These patients were attended to in healthcare centers of the Health Ministry of Health, Ecuador.

One of the major problems of elderly populations is the geographical access to health centers, due to loss of autonomy, difficult access to transport, financial difficulties, etc.... Not to mention those living in institutions: Your sampling is biased.

Regarding this observation, actually some participants were placed in a geriatric center, but as they had a primary health center as institution of reference, we associated the patient to their healthcare center. Now, this aspect has been included in the manuscript (Please, see lines 107-108).

  • Did you stratify by age group and gender? 

Thank you for this question, however, we did not stratify by age group or gender. We tried to get a representative sample through randomization.

  • Your inclusion-exclusion criteria are very limited

Thank you for your observation. We tried to follow similar criteria that were stablished in others studies. (Please, see references below):

  • Fernandez-Lazaro CI, García-González JM, Adams DP, Fernandez-Lazaro D, Mielgo-Ayuso J, Caballero-Garcia A, Moreno Racionero F, Córdova A, Miron-Canelo JA. Adherence to treatment and related factors among patients with chronic conditions in primary care: a cross-sectional study. BMC Fam Pract. 2019; 20(1):132. doi: 10.1186/s12875-019-1019-3. PMID: 31521114; PMCID: PMC6744672.
  • Tavares NU, Bertoldi AD, Mengue SS, Arrais PS, Luiza VL, Oliveira MA, Ramos LR, Farias MR, Pizzol TD. Factors associated with low adherence to medicine treatment for chronic diseases in Brazil. Rev Saude Publica. 2016;50(suppl 2):10s. doi: 10.1590/S1518-8787.2016050006150. PMID: 27982378; PMCID: PMC5157921.
  • Toledano-Toledano F, Contreras-Valdez JA. Validity and reliability of the Beck Depression Inventory II (BDI-II) in family caregivers of children with chronic diseases. PLoS One. 2018;13(11):e0206917. doi: 10.1371/journal.pone.0206917. PMID: 30485299; PMCID: PMC6261561.

  • You do not follow the STROBE guidelines process for writing and reading observational studies 

Thanks for this observation, we took into account this guideline when the manuscript was written. Now, we included sentence with this statement. Please, see line 81-82.

  • You write: "We used numerous recruitment strategies".

What does that mean?

We tried to reach all type of patients. Loja is a rural area, sometimes is difficult to reach some places, so we tried to recruit patients independently of their location. This aspect was already included in the manuscript (Please, see lines 102-105) 

  • Sample size

You will refer to diabetes. Why, when your research is multi-morbidity. In the calculation of your sample size, must intervene the range of age of your patients, sex, factors studied, SES, location, etc... The reasoning is based on a multistratified pathinder cluster approach as recommended by the WHO.

It is difficult to meet your specific objectives on such a small and unrepresentative base of subjects from the general population.

Thank you for your question.To calculate sample size, we considered the highest prevalence of the chronic diseases studied. In this case, this was the diabetes prevalence.

  • Ethical considerations

Please add the agreement number of the department

Regarding this information requested, was included in line 155.

  • You wrote: "Written consent was required for all participants".

In this age group, it is impossible to have written consent for the entire population: Subjects who are no  autonomous, subjects under guardianship, dependent subjects, Alzheimer, etc... Please can you specify?

Thank you for this observation. We changed text in the manuscript, and included this suggestion. Please, see lines 111-113.

  • Demographic Characteristics.

There are variables that are irrelevant or ethically questionable. Ex: Marital status Do you want to talk about one of the criteria that is part of the studies of seniors returning to isolation? If yes, there are 3-4 additional criteria to specify. If not, no interest.

We appreciate this comment, we included this variable because we wanted to draw sociodemographic characteristics of the sample. As you mentioned, with this question we tried to know if participants cohabited with a couple. We follow the structure used in other studies published in this journal: https://www.ncbi.nlm.nih.gov/pmc/articles/PMC5369087/pdf/ijerph-14-00251.pdf

  • Religion

Can you explain the logic of including this variable???? It is ethically questionable.

Following your recommendation, this variable has been deleted. Ecuador is a religious country and is very common to ask about this, but we agree with your observation and this does not add special information.  

The statistical analysis is well done.

Thank you for this comment.

  • Can you explain why you are talking about the average number of pathologies vs. Qualitative (ex: Number of chronic diseases)

Thank you very much for this comment. We have incorporated the following sentence in section “Materials and methods” (Please see lines 134-140): “Other studies have explored the association between number of chronic diseases and somatic, affective and cognitive symptoms (Jokela et al. 2019). Then, we hypothesized that increasing number of chronic diseases is most strongly associated with different signs and symptoms (pain, anxiety, Constipation, Difficulty sleeping, Dyspnoea, Fatigue, Hyporexia, Nausea, Sleepiness). In addition, due to small proportions of the different diseases studied in this study, chronic diseases were grouped into categories according to number of diseases to avoid to loss statistical power.”

Jokela, M., García-Velázquez, R., Airaksinen, J., Gluschkoff, K., Kivimäki, M., & Rosenström, T. (2019). Chronic diseases and social risk factors in relation to specific symptoms of depression: Evidence from the U.S. national health and nutrition examination surveys. Journal of Affective Disorders, 251(March), 242–247. https://doi.org/10.1016/j.jad.2019.03.074

  • "As a limitation we affirm these data are from only one region of Ecuador and may not be representative of the whole country."

Are the data representative of Loja?

As we show in table 1, sociodemographic characteristics are similar to the population living in the area. If we consult national statistics in Ecuador, we observe that are similar https://www.ecuadorencifras.gob.ec/wp-content/descargas/Manu-lateral/Resultados-provinciales/loja.pdf. However, we observe that our sample is formed by more men than women, when the situation is just the reverse (49,2% men vs 50,8%). We consider that this may be due to men decided to participate more than women (54% vs 46%).

Sample was selected by chronic process according to the main chronic diseases and data published about Loja on this website  (reference 8 in the paper) https://public.tableau.com/views/cronicas_2014_0/ANUARIO?:embed=y&:showVizHome=no&:display_count=y&:display_static_image=y

  • It is necessary to develop strategies to improve care provided to chronic patients in Ecuador in order to reduce associated morbidity.

Please can you in the discussion specify some tracks or recommendations made on this subject

Thank you for this observation. In order to answer your petition, we ask you to read in the discussion section lines 422-429. We suggest that patients should be included in decisions to embrace autonomy. Please, if you consider not enough this paragraph, let us know to improve it according your suggestions.

  • To decrease geriatric depression associated with chronicity, it is recommended to improve autonomy in daily life activities.

Unless I'm mistaken, this is the first time you've used the word "geriatric" in its generic sense. All other terms are related to "The Modified Geriatric Depression Scale of Yesavage". This brings me back to the comments about the profile of your study population.

Thanks for this observation. As we mentioned before, we stablished to include patients in the 60 years and over because WHO determines older population at this age.

"Targeted interventions to reduce the morbidity of chronic diseases in Ecuador are essential to improve health-related quality of life."

This term is used 4 times in this paper, 1 in summary and 2 in discussion. How does it relate to your results? Your introduction and discussion talk about GBD that are not quality of life.

Thank you for this comment. It is a fact that we did not measure quality of life so our manuscript is on focused on that. However, we assume that chronicity has an impact in the patients´ quality of life, as others authors suggest, so we made this relationship. We are aware that we have not measured this parameter, but we would like to do it in future research.  

  • You wrote: "At present, it has become evident that measuring years lived with disability (YLDs) is important for estimating the burden of non-lethal diseases and, consequently, for planning health service".

Yes, but concretely, how do you translate in your results the morbidity results observed in YLDS?

Thank you for this question. As we suggest in conclusions: Healthcare providers should establish strategies and interventions to reduce disabilities in older people (please see lines 447-448). At the same time, this is our future research in this field, to develop interventions with patients and familiar caregivers to reduce disabilities and improve their quality of life with workshops about cares.  

Round 2

Reviewer 1 Report

The manuscript still would require major revision of the statistical analysis, tables, and explanation of the results in order to be suitable for publication. The authors appear willing to make major changes, so my recommendations follow.

  1. Include the target population characteristics as a column in Table 1 so readers can see the compare the total sample to the target population.

Sample was selected by chronic process according to the main chronic diseases and data published about Loja on this website (reference 8 in the paper) https://public.tableau.com/views/cronicas_2014_0/ANUARIO?:embed=y&:showVi zHome=no&:display_count=y&:display_static_image=y

  1. I do not understand your statement. Was this simple random selection, or stratified by disease condition? In any case, put the explanation in the text.

  1. From what you wrote in the text, I cannot determine which GRANMO tab you used to calculate the sample size. You still do not say if you were trying to calculate sample size for a survey (i.e., meet a desired level of precision) or calculate group sizes for a comparison of two groups (i.e., have desired amount of power to detect a meaningful difference between the groups)? You wrote that you needed 80% power, but to detect what?

  1. Table 1, characteristics of subjects

State the numeric ranges for the BMI categories.

State the numeric ranges for the Katz Index categories.

State that in the KPS scale 100=normal and 0=dead.

Correction: in the Modified Geriatric Depression Scale, "Depression" should be defined as a score of 10 to 15.

  1. Table 2, comparison of means

Include a footnote explaining that IC95% is the 95% confidence interval for the difference (No-Yes) between the mean symptom score (or proportion with the symptom) among those without the disease (No) and among those with the disease (Yes).

Include IC95% for the difference in proportions with pain.

  1. Table 3, comparative analysis of clinimetric characteristics

State that in the KPS scale 100=normal and 0=dead.

For each clinimetric variable, state if you tested distributions among the categories (Chi-squared test), or equality of medians among the number of diseases groups (Kruskal-Wallis test). If you did Kruskal-Wallis, then state the medians, not just the category counts.

  1. Table 4: Associations between dependence level (Barthel Index) and different chronic diseases.

I agree that you could have used multinomial logistic regression, but I don't recommend it.

Why is "total dependence" in the header? This must be a mistake.

Why are crude ORs for visual disorders, pain, age, sex and BMI shown twice?

You should put crude ORs for number of chronic diseases into the same column as the other crude ORs. You can have another column for the adjusted ORs from Model 2.

The way you use BMI as an independent variable does not make sense. If you hypothesize that increased BMI is associated with increased odds of dependency then to maximize power you should treat BMI as an interval-ratio scale variable. If you hypothesize that abnormal BMI is associated with increased odds of dependency, then you should code BMI categories as three dummy binary variables (1=yes, 0=no) representing obesity, overweight and underweight. The fourth category (normal weight) would be implied, and represented by the state of all of the other three category variables being zero.

  1. Table 5. Associations between clinical and functional situation (KPS) and different chronic diseases or the number of chronic diseases.

Why are crude ORs for visual disorders, pain, age, sex and BMI shown twice?

You should put crude ORs for number of chronic disease categories into the same column as the other crude ORs. You can have another column for the adjusted ORs from Model 2.

The way you use BMI as an independent variable does not make sense. If you are trying to show that increased BMI is associated with increased odds of dependency then to maximize power you should treat BMI as an interval-ratio scale variable. If you are trying to show that abnormal BMI is associated with increased odds of dependency, then you should code BMI categories as three dummy binary variables (1=yes, 0=no) representing obesity, overweight and underweight. The fourth category (normal weight) would be implied, and represented by the state of all of the other three category variables being zero.

  1. Tables 6 and 7. Associations between depression (The Modified Geriatric Depression Scale of Yesavage) and different chronic conditions.

I agree that you may in principle use multinomial logistic regression, but I don't recommend it.

Tables 6 and 7 are very difficult to understand. Your garbled explanation of the findings (L312-319) suggests that even you don't understand what it shows. I don't blame you, neither do I.

You show odds ratios for "probable depression" (score 6-9) and "no depression" (score 0-5), but no odds ratios for "depression" (score 10-15). I think that you have designated "depression" as the reference category (OR=1 by definition), which may be technically acceptable, but makes no sense for purposes of interpretation. It would be more sensible to designate "no depression" as the reference category, and then the estimated odds ratios for "probable depression" and "depression" would be relative to the odds of "no depression".

The p-values for the models with one independent variable should be removed because they are not necessary and misleading. These p-values apply to the whole model and not to any specific crude odds ratio. In the text you claim a number of times a condition is associated with depression, but such claims need to be supported by the IC95% for the specific odds ratio.

The results would be easier to understand if you used logistic regression with a dichotomous outcome: "depression or probable depression" (score 6-15) or "no depression" (score 0-5), with "depression or probable depression" coded as "1" and "no depression" coded as "0". In categorical analyses, amalgamating categories can actually improve power by eliminating small cells and reducing degrees of freedom, and in your data set, few subjects had "depression" (score 10-15).

If you are really concerned about maximizing power, you could treat the Yesavage Geriatric Depression Scale (short form) as an interval-ratio scale variable, because it derives from a 15-item questionnaire, where all items are of equal weight (score 1 if answered in a manner suggestive of depression). Then you could use linear regression, and the estimated regression coefficients would represent the mean score change (points out of 15) associated with unit change in the independent variable.

Author Response

The manuscript still would require major revision of the statistical analysis, tables, and explanation of the results in order to be suitable for publication. The authors appear willing to make major changes, so my recommendations follow.

Thank you for your suggestions that improve our manuscript. We tried to address most requires, but if you consider that we would have to provide further information, we would be delighted to do it.

  1. Include the target population characteristics as a column in Table 1 so readers can see the compare the total sample to the target population.

Thanks for your comment but, at this stage, is not possible for us to address this request. Data provided in table 1 are not available in published statistics so we cannot include them. We hope you re-consider this request. We carried out this research to know more about this information due to lack of availability of this kind of data.

Sample was selected by chronic process according to the main chronic diseases and data published about Loja on this website (reference 8 in the paper) https://public.tableau.com/views/cronicas_2014_0/ANUARIO?:embed=y&:showVi zHome=no&:display_count=y&:display_static_image=y

  1. I do not understand your statement. Was this simple random selection, or stratified by disease condition? In any case, put the explanation in the text.

Thank you for this comment. We changed the text in the first version and included this information:

At the first stage, we contacted patients who attended an appointment in clinic to follow up their chronic problem, during home visits or geriatric institution visits (convenience sample). At the second stage, of those who met inclusion criteria, we randomized them using simple random sampling technique. Individuals were interviewed by trained staff who had no medical relationship with the patients. These patients were attended  in healthcare centers of the Health Ministry of Ecuador or they were living in a geriatric center associated with these healthcare center.

We would like to clarify that sample was not stratified by disease condition.

  1. From what you wrote in the text, I cannot determine which GRANMO tab you used to calculate the sample size. You still do not say if you were trying to calculate sample size for a survey (i.e., meet a desired level of precision) or calculate group sizes for a comparison of two groups (i.e., have desired amount of power to detect a meaningful difference between the groups)? You wrote that you needed 80% power, but to detect what?

Thank you very much for this comment. We made a mistake in writing, including the statistical power of 80%, while it had not been included for the sample calculation.

We have incorporated in section “Materials and Methods” (please, line 97) the following sentence: “Accepting a p value of < 0.05 and 5.0% precision level, 263 subjects were required.” (Sample included in our study is 283 participants)

  1. Table 1, characteristics of subjects

State the numeric ranges for the BMI categories.

Thank you for this recommendation that was done. Please, see table 1.

State the numeric ranges for the Katz Index categories.

Thank you for this observation. Katz Index was recoded into three categories: being independence (16 points), moderate dependence (8-15 points), and severe dependence (0-7 points). Please, see table 1.

State that in the KPS scale 100=normal and 0=dead.

Thank you for this comment. We have incorporated this information into table 1.

Correction: in the Modified Geriatric Depression Scale, "Depression" should be defined as a score of 10 to 15.

Thank you for this comment. We agree with you. We have modified this score in the table 1, the Modified Geriatric Depression Scale.

  1. Table 2, comparison of means

Include a footnote explaining that IC95% is the 95% confidence interval for the difference (No-Yes) between the mean symptom score (or proportion with the symptom) among those without the disease (No) and among those with the disease (Yes).

Include IC95% for the difference in proportions with pain.

Thank you very much for this comment. We have incorporated your suggestions into table 2. 

  1. Table 3, comparative analysis of clinimetric characteristics

State that in the KPS scale 100=normal and 0=dead.

For each clinimetric variable, state if you tested distributions among the categories (Chi-squared test), or equality of medians among the number of diseases groups (Kruskal-Wallis test). If you did Kruskal-Wallis, then state the medians, not just the category counts.

 Thank you very much for this comment. We have incorporated your suggestions into table 3.

  1. Table 4: Associations between dependence level (Barthel Index) and different chronic diseases.

I agree that you could have used multinomial logistic regression, but I don't recommend it.

Why is "total dependence" in the header? This must be a mistake.

Thank you very much for this comment. We have corrected the mistake.

Why are crude ORs for visual disorders, pain, age, sex and BMI shown twice?

You should put crude ORs for number of chronic diseases into the same column as the other crude ORs. You can have another column for the adjusted ORs from Model 2.

The way you use BMI as an independent variable does not make sense. If you hypothesize that increased BMI is associated with increased odds of dependency then to maximize power you should treat BMI as an interval-ratio scale variable. If you hypothesize that abnormal BMI is associated with increased odds of dependency, then you should code BMI categories as three dummy binary variables (1=yes, 0=no) representing obesity, overweight and underweight. The fourth category (normal weight) would be implied, and represented by the state of all of the other three category variables being zero.

Thank you for this comment. We have incorporated these changes into table 4. 

  1. Table 5. Associations between clinical and functional situation (KPS) and different chronic diseases or the number of chronic diseases.

Why are crude ORs for visual disorders, pain, age, sex and BMI shown twice?

You should put crude ORs for number of chronic disease categories into the same column as the other crude ORs. You can have another column for the adjusted ORs from Model 2.

The way you use BMI as an independent variable does not make sense. If you are trying to show that increased BMI is associated with increased odds of dependency then to maximize power you should treat BMI as an interval-ratio scale variable. If you are trying to show that abnormal BMI is associated with increased odds of dependency, then you should code BMI categories as three dummy binary variables (1=yes, 0=no) representing obesity, overweight and underweight. The fourth category (normal weight) would be implied, and represented by the state of all of the other three category variables being zero.

Thank you very much for this comment. We have incorporated these changes into Table 5.

  1. Tables 6 and 7. Associations between depression (The Modified Geriatric Depression Scale of Yesavage) and different chronic conditions.

I agree that you may in principle use multinomial logistic regression, but I don't recommend it.

Tables 6 and 7 are very difficult to understand. Your garbled explanation of the findings (L312-319) suggests that even you don't understand what it shows. I don't blame you, neither do I.

You show odds ratios for "probable depression" (score 6-9) and "no depression" (score 0-5), but no odds ratios for "depression" (score 10-15). I think that you have designated "depression" as the reference category (OR=1 by definition), which may be technically acceptable, but makes no sense for purposes of interpretation. It would be more sensible to designate "no depression" as the reference category, and then the estimated odds ratios for "probable depression" and "depression" would be relative to the odds of "no depression".

The p-values for the models with one independent variable should be removed because they are not necessary and misleading. These p-values apply to the whole model and not to any specific crude odds ratio. In the text you claim a number of times a condition is associated with depression, but such claims need to be supported by the IC95% for the specific odds ratio.

The results would be easier to understand if you used logistic regression with a dichotomous outcome: "depression or probable depression" (score 6-15) or "no depression" (score 0-5), with "depression or probable depression" coded as "1" and "no depression" coded as "0". In categorical analyses, amalgamating categories can actually improve power by eliminating small cells and reducing degrees of freedom, and in your data set, few subjects had "depression" (score 10-15).

If you are really concerned about maximizing power, you could treat the Yesavage Geriatric Depression Scale (short form) as an interval-ratio scale variable, because it derives from a 15-item questionnaire, where all items are of equal weight (score 1 if answered in a manner suggestive of depression). Then you could use linear regression, and the estimated regression coefficients would represent the mean score change (points out of 15) associated with unit change in the independent variable.

Thank you very much for this comment. We have carried out a logistic regression with a dichotomous outcome: “depression or probable depression” (score 6-15) or "no depression" (score 0-5), with "depression or probable depression" coded as "1" and "no depression" coded as "0". Tables 6 and 7 have been unified as Table 6.

Author Response

Thank you for your support to publish this manuscript. Kind Regards, 

Round 3

Reviewer 1 Report

The third version of the manuscript still contains serious errors in the statistical calculations. These require correction if the manuscript is to be fit for publication.

L125-L127 and Table 1

Is this 16-point scale called the "Katz Index"? Please provide a reference. As far as I know, the Katz Index of Activites of Daily Living is a 6-point numeric ordinal scale.

Table 2

Your 95% confidence intervals for the difference in proportions with pain are incorrect. In the attached spreadsheet I used a simple method to estimate the confidence interval (normal approximation of the binomial distribution)

https://www.dummies.com/education/math/statistics/how-to-estimate-the-difference-between-two-proportions/.

Your confidence intervals are not even close to what I calculate. I think you mistakenly calculated the 95% confidence interval for the ratio of two proportions.

You have incorrectly indicated that the disparity in proportion with pain among those with arterial hypertension compared to the proportion among those without arterial hypertension is statistically significant (p<0.05). This is not consistent with the 95% confidence interval, either for the difference of the two proportions, or the ratio of the two proportions.

Table 3, footnote

You mistakenly wrote, "We show the average values and standard deviations in brackets when a Chi-squared test was applied, or frequency and percentages in brackets when a Kruskal Wallis test was applied." You should say, "We show the average values and standard deviations in brackets when a Kruskal Wallis test was applied, or frequency and percentages in brackets when a Chi-squared test was applied."

Tables 4 and 5:

The 95% confidence limits of the odds ratios have not been correctly calculated and reported.

The point estimate of the odds ratio must be within the 95% confidence interval.

An odds ratio must be greater than zero. It is not possible for the lower 95% confidence limit of the odds ratio to be negative (less than zero).

If the 95% confidence interval for the odds ratio includes one, then it is not possible for the odds ratio to be statistically significantly different from one (p<0.05).

I think you mistakenly reported the 95% confidence limits of the regression coefficient. The odds ratio is the exponentiation of the regression coefficient. The 95% confidence limits of the odds ratio are obtained by the exponentiation of the 95% confidence limits of the regression coefficient.

Table 6:

The way you have coded Katz Index (dependence) is confusing. If you are trying to show that dependence is associated with increased odds of depression, then you should code dependence as two dummy binary variables (1=yes, 0=no) representing severe dependence and moderate dependence. The third category (independence) would be implied, and represented by the state of the other two category variables both being zero.

Author Response

The third version of the manuscript still contains serious errors in the statistical calculations. These require correction if the manuscript is to be fit for publication.

L125-L127 and Table 1

Is this 16-point scale called the "Katz Index"? Please provide a reference. As far as I know, the Katz Index of Activites of Daily Living is a 6-point numeric ordinal scale.

Thank you for this comment. We have provided several references and have included the following sentence in section “Materials and Methods” (lines 127-137): “In accordance with Clinical Guides Geronto - Geriatric Primary Health Care for the Elderly published by Ecuador´s Ministry of Public Health [16], the Katz Index modified consists of 8 items. Each item, such as the original scale, is scored on three levels from 0 (dependent) to 2 (independent). The value 1 means moderate dependence. Then, the total score can range from 0 to 16. Several forms of interpretation have been proposed [17-26]. For instance, Palomo & Gervás [27] used this index from Cruz [28] considering “independent” to those who scored 2 in each items, “dependent” to those who scored 0 in each items, and “partially dependent” to the rest. Given the consense gap regarding the classification of this index modified used in Ecuador and since the guides used do not specify any global classification of this scale, in this study the Katz Index was recoded into three categories: independence (16 points), moderate dependence (8-15 points) and severe dependence (0-7 points).

Table 2

Your 95% confidence intervals for the difference in proportions with pain are incorrect. In the attached spreadsheet I used a simple method to estimate the confidence interval (normal approximation of the binomial distribution)

https://www.dummies.com/education/math/statistics/how-to-estimate-the-difference-between-two-proportions/.

Your confidence intervals are not even close to what I calculate. I think you mistakenly calculated the 95% confidence interval for the ratio of two proportions.

You have incorrectly indicated that the disparity in proportion with pain among those with arterial hypertension compared to the proportion among those without arterial hypertension is statistically significant (p<0.05). This is not consistent with the 95% confidence interval, either for the difference of the two proportions, or the ratio of the two proportions.

Thank you very much for this comment. We have corrected the 95% confidence intervals for the difference in proportions with pain.

Table 3, footnote

You mistakenly wrote, "We show the average values and standard deviations in brackets when a Chi-squared test was applied, or frequency and percentages in brackets when a Kruskal Wallis test was applied." You should say, "We show the average values and standard deviations in brackets when a Kruskal Wallis test was applied, or frequency and percentages in brackets when a Chi-squared test was applied."

Thank you for this comment. We are sorry for this mistake. We have modified the footnote (Table 3).

Tables 4 and 5:

The 95% confidence limits of the odds ratios have not been correctly calculated and reported.

The point estimate of the odds ratio must be within the 95% confidence interval.

An odds ratio must be greater than zero. It is not possible for the lower 95% confidence limit of the odds ratio to be negative (less than zero).

If the 95% confidence interval for the odds ratio includes one, then it is not possible for the odds ratio to be statistically significantly different from one (p<0.05).

I think you mistakenly reported the 95% confidence limits of the regression coefficient. The odds ratio is the exponentiation of the regression coefficient. The 95% confidence limits of the odds ratio are obtained by the exponentiation of the 95% confidence limits of the regression coefficient.

Thank you very much for this comment. We made a mistake by incorporating the data into the table. We have changed the 95% confidence limits in tables 4 and 5.

 Table 6:

The way you have coded Katz Index (dependence) is confusing. If you are trying to show that dependence is associated with increased odds of depression, then you should code dependence as two dummy binary variables (1=yes, 0=no) representing severe dependence and moderate dependence. The third category (independence) would be implied, and represented by the state of the other two category variables both being zero.

Thank you very much for this comment. We have corrected the Katz Index in table 6.
